# Fault Detection and Diagnosis for Liquid Rocket Engines Based on Long Short-Term Memory and Generative Adversarial Networks

**Lingzhi Deng, Yuqiang Cheng * and Yehui Shi**

College of Aerospace Science and Engineering, National University of Defense Technology, 109 Deya Road, Changsha 410073, China; denglingzhi@nudt.edu.cn (L.D.); shiyehui@nudt.edu.cn (Y.S.)
* Correspondence: cheng_yuqiang@163.com

**Abstract:** The development of health monitoring technology for liquid rocket engines (LREs) can effectively improve the safety and reliability of launch vehicles, which has important theoretical and engineering significance. Therefore, we propose a fault detection and diagnosis (FDD) method for a large LOX/kerosene rocket engine based on long short-term memory (LSTM) and generative adversarial networks (GANs). Specifically, we first modeled a large LOX/kerosene rocket engine using MATLAB/Simulink and simulated the engine's normal and fault operation states involving various startup and steady-state stages utilizing fault injection. Second, we created an LSTM-GAN model trained with normal operating data using LSTM as the generator and a multilayer perceptron (MLP) as the discriminator. Third, the test data were input into the discriminator to obtain the discrimination results and realize fault detection. Finally, the test data were input into the generator to obtain the predicted samples and calculate the absolute error between the predicted and the real value of each parameter. Then the fault diagnosis index, standardized absolute error (SAE), was constructed. SAE was analyzed to realize fault diagnosis. The simulated results highlight that the proposed method effectively detects faults in the startup and steady-state processes, and diagnoses the faults in the steady-state process without missing an alarm or being affected by false alarms. Compared with the conventional redline cut-off system (RCS), adaptive threshold algorithm (ATA), and support vector machine (SVM), the fault detection process of LSTM-GAN is more concise and more timely.

**Keywords:** liquid rocket engine; LSTM; GAN; fault detection; fault diagnosis; SAE





## 1. Introduction

The reliability of a liquid rocket engine (LRE), as the central power unit of a launch vehicle, is critical to the success of a space mission. However, complex LRE systems and extreme working environments increase the LRE's failure probability, which develops rapidly and has strong destructive power, often leading to severe consequences, including the loss of the vehicle [1]. Nevertheless, health monitoring technology, whose core and foundation are fault detection and diagnosis (FDD), can improve the LRE's safety and reliability [2].

Since the 1970s, the United States has employed the red-line system [3], under the guidance of the SSME [4], space launch [5], and integrated space launch [6] programs, to reduce the impact of space shuttle main engine (SSME) failure and improve its reliability and safety. The red-line system comprises various systems such as the System for Anomaly and Failure Detection (SAFD) [7], Health Monitoring System (HMS) [3], Intelligent Control System (ICS) [8], Health Management System for Rocket Engines (HMSRE) [9], Integrated Vehicle Health Management System (IVHMS) [10], and the Advanced Health Management System (AHMS) [11,12]. Furthermore, it involves the engine data interpretation system [13], auto-

mated propulsion data screening demonstration system [14], and the post-test diagnostic expert systems [15] for liquid rocket engine test run/post-flight data analysis.

Additionally, researchers have applied a series of FDD algorithms based on signal processing to LRE data. For example, ref. [16] improved the adaptive threshold algorithm and developed a real-time fault detection and alarm system for engine ground tests based on lab windows/cvi. In [17], the authors proposed an adaptive Gaussian threshold model for online detection of turbopump vibration signals utilizing time-domain characteristics (root mean square, kurtosis factor, and peak factor) of LRE turbopump normal data that conform to a Gaussian distribution. The work of [18] developed an FDD algorithm based on a nonlinear Kalman filter for the transient process of an open cycle LRE.

In recent years, deep learning and other machine learning methods have developed rapidly and have been widely used in the FDD field. For instance, ref. [19] applied a quantum genetic algorithm (QGA) for the parameter training and optimization of back propagation (BP) neural networks and proposed a fault detection method for an LRE based on QGA-BP. In [20], the authors proposed a tool wear process condition monitoring method for aerospace manufacturing based on a convolutional neural network (CNN) to identify intermediate abnormal states in multi-stage processes. This method's feasibility was verified on an open-source data set. The work of [21] applied an inception-CNN to the fault detection of aero-engine sensors and verified the effectiveness and feasibility of this method in terms of two aspects: the typical sensor fault detection effect and the fault detection and isolation process. Ref. [22] proposed a text-based fault diagnosis model that utilized word2vec to extract text feature vectors and employed a stack-based integrated learning scheme for classification. The performance was verified on real aircraft fault text data sets. A convolutional auto-encoder method that extracts parameter features of the LREs and a one-class supported vector machine to realize steady-state fault detection for LREs have also been proposed [23]. However, current methods rely on many fault samples and cannot model the entire variable thrust LRE process well. Furthermore, multiple models are required for segmented detection, which is a complex data processing procedure.

As a representative of temporal data processing methods in deep learning methods, the LSTM network is widely used in FDD utilizing temporal data. For example, ref. [24] solved the imbalance problem using a GAN to synthesize the fault data, which allows the LSTM to learn the time correlation of data and classify the pipeline status to predict leakage. Additionally, ref. [25] conducted FDD for LRE startup transience, realized LRE fault detection using LSTM, and performed fault diagnosis through CNN-LSTM. This method was evaluated on the actual LRE test data set. In [26], the authors introduced a discrete wavelet transform into an LSTM model for multi-sensor fault diagnosis, and [27] applied LSTM to the fault diagnosis of electric vehicles and tested their scheme in simulated and practical data of electric vehicles, demonstrating an appealing accuracy that outperformed other methods. LSTM has previously been combined with a support vector data description algorithm [28] to dynamically adjust the fault residual according to the absolute difference between the predicted and the actual value. Such a system can realize the real-time fault detection of heating, ventilation, and air conditioning systems. In [29], the authors adaptively applied the chirp mode decomposition, Gini index fusion, and Aquila optimizer to LSTM for intelligent diagnosis of bearing composite faults, [30] applied LSTM to fault detection and identification of four rotor blades, and [31] utilized LSTM for fault detection of a high voltage direct current system, with an accuracy of 100%.

LRE has many failure modes involving limited and incomplete failure data. It is more appropriate to use an unsupervised learning method to realize LRE FDD, and this method can detect new faults. When the learning method is unsupervised, the most common practice of LSTM in applying FDD is to delimit the threshold according to the predicted value or actual value. When the predicted value or actual value exceeds the threshold, it is considered that an anomaly has occurred. However, this process has some subjective factors, and the threshold is not always reasonable. In order to avoid this problem and

simplify the fault detection process, we introduce LSTM into GAN and use the classification function of GAN to realize fault detection.

This paper proposes an FDD method based on LSTM and GAN for a large LOX/kerosene rocket engine. In our method, first, a large LOX/kerosene rocket engine is modeled using MATLAB/Simulink, and the engine's normal operation state is simulated. The fault operation states of various startup and steady-state stages are simulated by fault injection, which is achieved by modifying the failure factor. Second, the LSTM-GAN model uses LSTM as the generator and MLP as the discriminator, while the model is trained with normal data. Third, the test data are input into the discriminator to obtain the discrimination results and realize the fault detection of the whole process. Finally, the test data are input into the generator to obtain the predicted samples, and the absolute error between the predicted and the real value of each parameter is calculated. This is then standardized, and referred to as the standardized absolute error (SAE). SAE is analyzed to realize the steady-state process fault diagnosis of the engine. Simulation data verify the effectiveness and timeliness of the proposed method, which has three main points of innovation and advantages.

(1) The generator of GAN is constructed by LSTM, and the prediction function of LSTM for time series data and the classification function of GAN are used at the same time. The whole process fault detection of an LRE can be realized using only one model. Compared with RCS, ATA, and SVM methods, which need to detect data segmentation, the proposed method simplifies the fault detection process and has apparent advantages in terms of timeliness.
(2) The fault diagnosis index, SAE, is constructed. The diagnosis process is simple and the result is reliable.
(3) Only the normal data are used to train LSTM-GAN, so the FDD of new faults can be realized.

The remainder of this paper is organized as follows: Section 2 introduces the engine simulation and fault injection process. Section 3 presents the construction of the LSTM-GAN model. Section 4 discusses the validation of the proposed method using simulation data. Section 5 concludes this paper.

## 2. Simulation Process of an LRE

Figure 1 shows the functional block diagram of the simulation process of an LRE. The real-time simulation platform uses MATLAB/Simulink for modeling and the VxWorks operating system for the closed-loop simulation iteration process. This process first employs modular modeling and simulation, relying on MATLAB/Simulink to establish the model's components under different states, such as the liquid rocket engine combustion chamber, fuel preloading turbine model, gas turbine model, pump model, and thermal component model. The components are connected according to the correlation between the engine components, and an off-line simulation ensures the model's feasibility and accuracy. Then, according to the links and RT hardware driver module requirements of the general real-time simulation platform, I/O hardware interfaces such as the channel initialization module and the lower computer synchronization module are split and added to meet the hardware/software real-time communication requirements in the loop simulation platform. Through the main control software, RT-sim, we set the attribute parameters of the real-time computer simulator, deploy the corresponding target machine, and select and save the variable parameters that need to be monitored, recorded, and edited. Finally, the real-time simulation system realizes the online monitoring and modification of the variable parameters through the monitoring panel to convert the working conditions and fault injection. The RT hardware and I/O hardware used here comes from the company Linkstech, Changping District, Beijing, China.

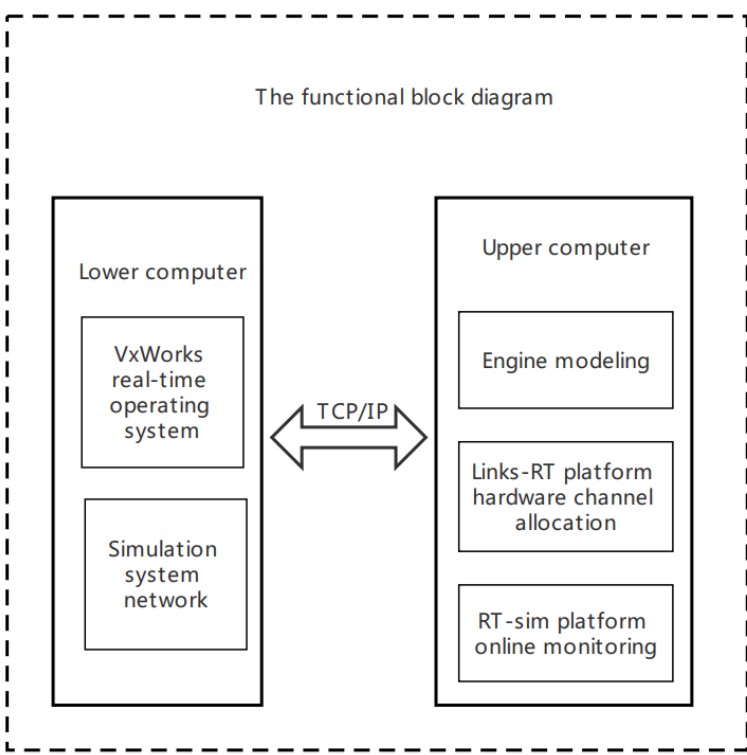

**Figure 1.** The functional block diagram.

Figure 2 depicts the structure diagram of the LRE simulated in this research involving the following engine working process. Before starting, the kerosene circuit of the generator and thrust chamber are first subjected to a forced purge. Then, the high-pressure helium gas squeezes the kerosene in the starting box to break the ignition tube diaphragm, and the ignition agent is divided into two to fill the generator and the thrust chamber ignition path. Then, the main liquid oxygen valve is opened, and the engine's liquid oxygen enters the generator. When the ignition agent in the ignition circuit of the thrust chamber is filled to the required position, the generator's fuel valve opens, and the ignition agent enters the generator and ignites with the liquid oxygen entered in advance. When the ignition starts, the fuel flows into the generator starting the flow, the generator's component ratio drops rapidly, and the temperature of the oxygen-rich gas rises, driving the main turbine to rotate. As the throttle valve in the main fuel circuit is in a small flow state, the flow of the coal oil pump is almost zero, the flow and head of the oxygen main pump increase, the power required by the pump increases, the component ratio of the generator gradually increases, and the power of the main turbine also increases. Then, the main turbine's power and the pump's required power gradually tend to balance. At this time, the ignition fluid enters the thrust chamber and burns with the fuel gas entering the thrust chamber after driving the main turbine to rotate. Before starting, the kerosene is filled in front of the thrust chamber fuel main valve through the thrust chamber nozzle cooling channel. When the valve opens, the throttle valve is in a small flow state, and kerosene flows to the injector along the cooling channel of the thrust chamber. As the generator pressure increases and the component ratio gradually approaches the rated value, the turbine pump speed increases, and the pump back-pressure and propellant flow increase. When the outlet pressure of the primary fuel pump reaches the given value, the hydraulic relay starts to work, the flow regulator turns to primary, the fuel flow of the control generator gradually increases, the engine changes to a high working condition, and the engine parameters continue to rise. Kerosene enters the thrust chamber to realize supplementary combustion and burns with oxygen-enriched gas. The engine's operation shifts to the main stage and shuts down after completing its task. However, before the shutdown, the engine turns to the final working condition and closes the generator's fuel valve to complete the shutdown process.

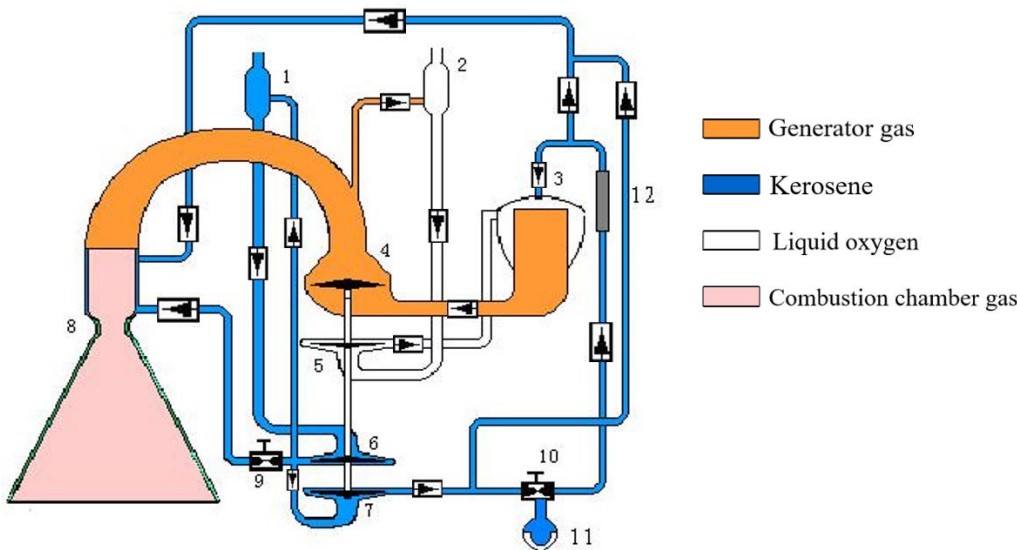

**Figure 2.** Diagram of the LRE structure. (1. Fuel pre-pressure turbine pump; 2. oxidant pre-pressure turbine pump; 3. gas generator; 4. main turbine; 5. oxidant pump; 6. fuel primary pump; 7. fuel secondary pump; 8. combustion chamber; 9. throttle valve; 10. flow regulator; 11. starting box; 12. ignition duct).

We obtain nine normal operation status data through the simulation system, three startup process fault data, and four steady-state stage fault data, with the specific operating conditions reported in Table 1. The nine normal operation states involve multiple but different operating condition conversions and slightly different model parameters. Steady-state faults are realized by modifying the model parameters during simulation, and the startup faults are realized by changing parameters before simulation. The model parameters of the fault simulation scenario are 24-channel sensor signals from 10 critical components of the LRE, having a sampling rate of 500 Hz (Table 2). These parameters are different from the normal operation parameters.

**Table 1.** The operating conditions of the LRE.

| Operating Conditions | Test Number | Operating Condition Conversions/Fault Modes |
|---|---|---|
| Normal | 01, 02, 03, 04, 09 | Low operating condition–rated operating condition–high operating condition–rated operating condition–final operating condition |
| | 05, 06, 07, 08 | Low operating condition–rated operating condition–final operating condition |
| Steady-state faults | 11 | Combustion chamber throat ablation |
| | 12 | Stuck bearing |
| | 13 | Cavitation of oxidant pump |
| | 14 | Blocked pipeline in front of oxidant pump |
| Start-up transient faults | 15 | Blocked pipeline in front of oxidant pump |
| | 16 | Main turbine rotor damage |
| | 17 | Stuck bearing |

**Table 2.** Components of the LRE, parameters, and acronyms.

| Components | Parameters | Acronyms | Components | Parameters | Acronyms |
|---|---|---|---|---|---|
| Combustion chamber | Inlet fuel flow<br>Pressure<br>Temperature | qc<br>pc<br>Tc | Primary fuel pump | Inlet pressure<br>Outlet pressure<br>Flow | pifp1<br>pofp1<br>qpfp1 |
| Gas generator | Inlet fuel flow<br>Inlet oxidizer flow<br>Pressure | qfg<br>qog<br>pg | Oxidizer<br>pre-pressure pump | Inlet pressure<br>Outlet pressure<br>Flow | piopp<br>poopp<br>qopp |
| Main turbine | Torque<br>Rotation rate | Mt<br>Nt | Secondary fuel<br>pump | Inlet pressure<br>Outlet pressure | pifp2<br>pofp2 |
| Fuel pre-pressure pump | Inlet pressure<br>Outlet pressure<br>Flow | pifp<br>pofp<br>qfp | Oxidizer pump | Inlet pressure<br>Outlet pressure<br>Flow | piop<br>poop<br>qop |
| Fuel preload turbine | Rotation rate | nft | Oxidizer preload<br>turbine | Rotation rate | not |

## 3. Introduction to LSTM-GAN

### 3.1. Introduction to LSTM Networks

#### 3.1.1. The Basic Structure

A recurrent neural network (RNN) is an artificial neural network (ANN) with a time memory function. An RNN affects the output using memory cells that store past inputs. However, the long-term dependency significantly restricts the RNN's performance. To address this problem, LSTM introduces an input gate, output gate, and a forgetting gate into the RNN (Figure 3).

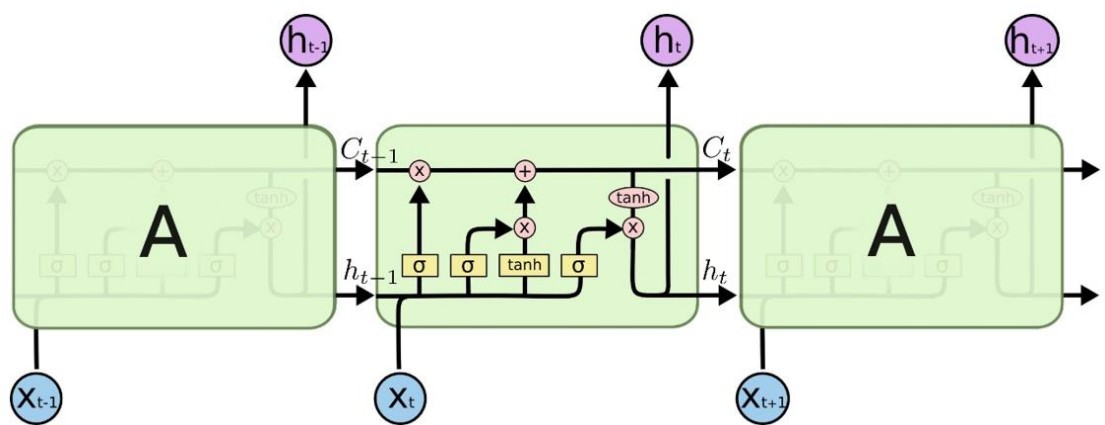

**Figure 3.** The LSTM structure.

The equations describing a basic unit of LSTM at a certain time *t* are:

$$f_t = \sigma(W_{hf}h_{t-1} + W_{xf}x_t + b_f) \tag{1}$$

$$i_t = \sigma(W_{hi}h_{t-1} + W_{xi}x_t + b_i) \tag{2}$$

$${c_t}' = \tanh(W_{hc}h_{t-1} + W_{xc}x_t + b_c) \tag{3}$$

$$o_t = \sigma(W_{ho}h_{t-1} + W_{xo}x_t + b_o) \tag{4}$$

$$c_t = f_t c_{t-1} + i_t {c_t}' \tag{5}$$

$$h_t = o_t \tanh(c_t) \tag{6}$$

where $x$, $W$, and $b$ are the input, weight, and bias, respectively. $\sigma$ is a sigmoid function and restricts the output range to (0, 1), calculated by the function $\sigma(x) = 1/(1 + e^{-x})$. tanh is

a hyperbolic tangent function and restricts the output range to $(-1, 1)$, calculated by the function $\tanh(x) = (e^x - e^{-x})/(e^x + e^{-x})$. $i$ is the input gate and determines whether the current input is added to the storage unit and which values to update. $f$ is the forgetting gate and determines whether the values in the memory unit remain unchanged, decrease, or are reset. $o$ is the output gate that determines the output through the input and the memory unit values. $C'$ controls the input to be updated. $c$ is the cell condition. $h$ is the hidden condition and is usually the output.

### 3.1.2. The Input and Output

Figure 3 shows that the LSTM's input and output are temporal signals, including spatial and time dimensional information. In this research, the spatial information involves 24-channel sensor signals, and the temporal information is obtained by oversampling the data in the time domain. The FDD task ensures that only the data are obtained according to the input signals at a specific time. Therefore, the output's time dimension is set to one. The dimension of the input signals is analyzed in Section 4.

### 3.2. Introduction to GAN

GAN is a framework for estimating the generation model through a confrontation process [32]. It can learn the distribution of actual samples without a priori probability modeling. GAN comprises a generator and a discriminator model relying on neural networks. The generator maps the input noise signals to the sample space through the neural networks and outputs the fake data to be as true as possible. The discriminator judges whether the sample originates from the generator's fake samples or the real samples in the real data distribution. During GAN training, both models are optimized alternately to achieve Nash equilibrium. Finally, the samples generated by the generator conform to the real sample probability distribution. After completing the training process, the generator estimates the original data's probability distribution well. Figure 4 illustrates the classic GAN training process.

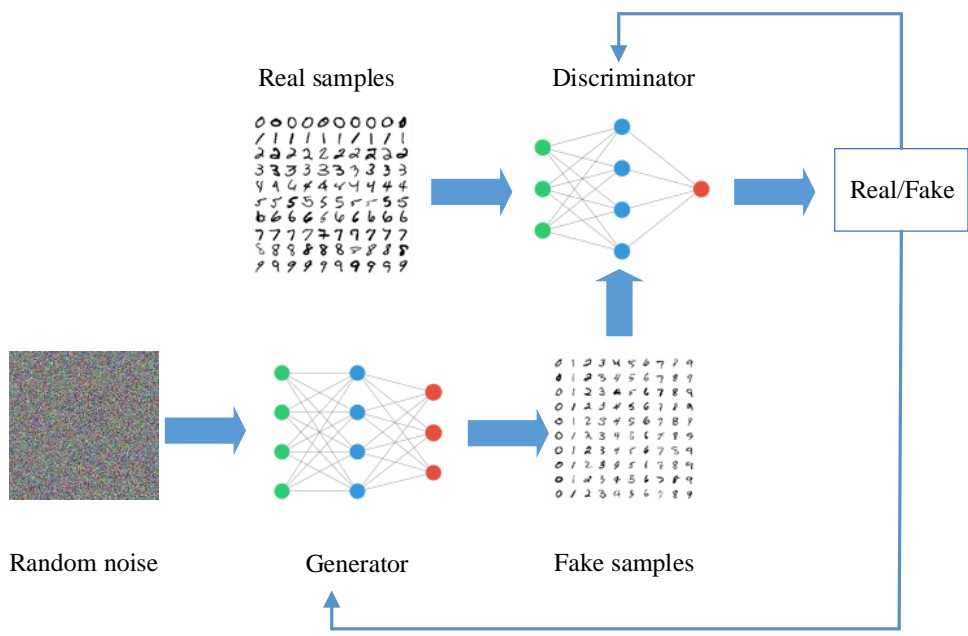

**Figure 4.** The classic GAN training process.

### 3.3. Architecture of LSTM-GAN

#### 3.3.1. The Basic Structure

The traditional GAN cannot efficiently learn the data's time correlation, so we utilize an LSTM-based architecture to build the generator. Given that the discriminator judges

whether the input sample is from the real sample distribution, the discriminator relies on an MLP. The input parameters of the generator are the LRE's 24 parameters in a certain period. Furthermore, the generator has two hidden layers, where each layer has 128 LSTM units. To prevent overfitting, both layers have dropout layers, with a dropout probability of 0.3, and the output parameters are the LRE's 24 parameters at the current time. The discriminator's input parameters are the LRE's 24 parameters. The discriminator involves two hidden layers comprising 64 and 128 neurons, respectively, and LeakyReLU is selected as the activation function. The output result is the fault detection result. The activation functions of the output layers of the generator and the discriminator are sigmoid functions. Finally, the training process utilizes the Adam optimizer with an initial learning rate of 0.0005 and a batch size of 1024.

### 3.3.2. Loss Function

Regression problems, to ensure the error between the predicted value and the real value is as small as possible, typically employ the mean square error (*MSE*) loss function, calculated as:

$$MSE = \sum_{i=1}^{n} (x - x_r)^2 \tag{7}$$

where $x$ is the input sample and $x_r$ is the reconstructed sample. Accordingly, the binary cross-entropy (*BCE*) loss function is typically used in a two-classification problem. The BCE formula is:

$$BCE = -(xlog(p) + (1 - x)log(1 - p)) \tag{8}$$

where $x$ is the real label and $p$ is the predicted label.

The generator built by LSTM solves the regression problem, so the *MSE* loss function is selected. In addition, since the generator needs to consider the discriminator's result when updating the parameters, the generator's loss function is obtained by adding the two parts to obtain the average value, i.e.,

$$L_g = (MSE_1 + MSE_2)/2 \tag{9}$$

where $MSE_1$ is calculated by the generator's parameters and the actual output parameters, and $MSE_2$ relies on the discriminator's result and the actual label. To preserve consistency with the generator, we select the *MSE* loss function for the discriminator.

## 4. Experiments and Analysis

### 4.1. Data Preprocessing

This research does not consider the shutdown process, so the corresponding data are removed. The original data samples comprise four parameter types, i.e., pressure, temperature, mass flow, and rotating speed, ranging from $10^1$ to $10^4$. Furthermore, the data are normalized to narrow the data distribution range to [0, 1] and ease the neural network's training process. The calculation equation is:

$$y = \frac{x - x_{min}}{x_{max} - x_{min}} \tag{10}$$

where $x_{max}$ and $x_{min}$ are a parameter's maximum and minimum values, respectively. In order to avoid information about the test set being introduced into the training process, the maximum and minimum values of the training set are used to preprocess all samples in the training and test sets.

The input data of the generator are time series data. Thus, data overlap in time, and the sampling step is set to one, as illustrated in Figure 5. In order to explore the impact of the sample size on the model's performance, we set five window lengths having a sample size of $10 \times 24$, $20 \times 24$, $30 \times 24$, $40 \times 24$, and $50 \times 24$. The input data of the discriminator are the samples of the following input data of the generator, so no additional processing is required.

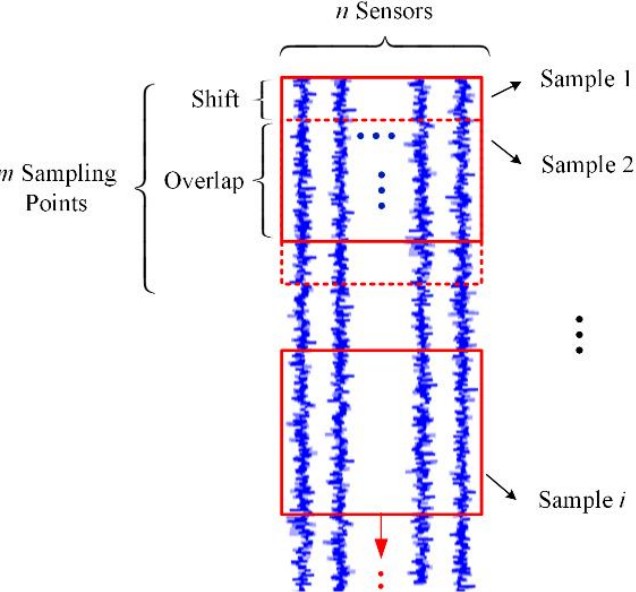

**Figure 5.** Overlapping sampling process.

Given that time series data are considered, ensuring integrity in time series is necessary. Therefore, test data 01–08 comprise the training set, and 09 and 11–16 are the test set. The training set is utilized for training LSTM-GAN and the test set is used to verify whether the method has false or missing alarms, and the timeliness of the method.

### 4.2. Model Training

The deep learning environment of this research was based on Python 3.8.12 and PyTorch 1.10.0. The operating environment was Microsoft Windows 11, and processing was performed on an NVIDIA GeForce GTX 1650 with 4 GB VRAM. All experiments were completed on the jupyter notebook. Considering the LSTM-GAN with a sample size of $10 \times 24$ as an example, the training data were input into the LSTM-GAN-10 for training, where the training process is illustrated in Figure 6. After about 2600 iterations, the losses of the generator and the discriminator stabilized and were close to zero, reaching an ideal state.

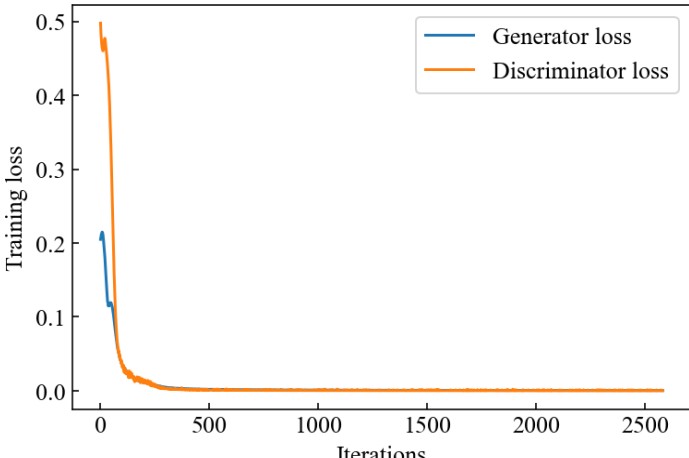

**Figure 6.** The training process of LSTM-GAN with a sample size of $10 \times 24$.

### 4.3. Evaluating Indicator

In the FDD field, the precision, recall, and F1 metrics are typically used to measure the performance of the evaluated methods [33]. Considering FDD for LREs, finding anomalies

and faults in operation as soon as possible and avoiding false alarms is essential [23]. Nevertheless, due to abnormal sensors and significant transient parameter fluctuation during the LRE operation, outliers that deviate significantly from the normal value may occasionally appear in LRE commissioning data. When detecting outliers, it does not mean a fault has occurred. Therefore, a continuity criterion is usually introduced that considers the fault and issues an alarm after three consecutive abnormal detections. This strategy improved the method's reliability and reduced the false alarm rate.

Furthermore, this paper proposes three evaluation indicators to evaluate the proposed method's performance. The two qualitative indicators are the occurrences of false alarms and missing alarms, and the quantitative indicator is the alarm time of the fault samples, for which, the earlier, the better. We expect the proposed method will not issue false or missing alarms, and can realize early detection of faults from the fault test data.

### 4.4. Fault Detection Based on Discriminator

The discriminator only learned the sample distribution of normal data, so the discrimination result for normal samples should be close to 1, and the discrimination result for abnormal samples should be close to 0. Fault detection is carried out according to this principle. The test data are input to the discriminator, which outputs the result. If the discrimination result exceeds 0.5, the output is set to one, indicating normal conditions. Otherwise, the output is set to zero, indicating abnormal conditions. When three consecutive abnormal samples occur, it is considered that a fault has occurred, and an alarm is sent. The difference between the alarm and fault injection times, called the detection delay, is calculated and recorded.

The LSTM-GAN having different sample sizes presented no false alarms on the test set, proving that the discriminator has successfully learned the entire LRE operation and does not give any false alarms during the operating condition conversion. The corresponding performance on the six fault test sets is reported in Table 3, which highlights that the LSTM-GAN having a sample size of $30 \times 24$, $40 \times 24$, and $50 \times 24$ has missing alarms. The LSTM-GAN having a sample size of $10 \times 24$ and $20 \times 24$ can detect the injected fault in time, proving that the discriminator accurately judges the consistency between the test and the training samples. Hence, the results demonstrate that the developed fault detection method meets the requirements of timeliness and effectiveness.

**Table 3.** Detection delay of LSTM-GAN having different sample sizes.

| Test No. | $10 \times 24$/s | $20 \times 24$/s | $30 \times 24$/s | $40 \times 24$/s | $50 \times 24$/s |
|---|---|---|---|---|---|
| 11 | 0.006 | 0.100 | 0.020 | 0.006 | 0.040 |
| 12 | 0.020 | 0.012 | / | / | / |
| 13 | 0.010 | 0.016 | 0.204 | 0.006 | 0.166 |
| 14 | 0.006 | 0.012 | 0.010 | 0.036 | 0.010 |
| 15 | 1.136 | 1.390 | 1.152 | 1.374 | 1.350 |
| 16 | 1.554 | 1.164 | / | 2.024 | 3.286 |
| 17 | 1.552 | 2.336 | / | 1.668 | 3.118 |

'/' means missing alarm.

In most cases, the LSTM-GAN having a sample size of $10 \times 24$ performs well. Thus, we compared it with RCS, ATA, and SVM, as shown in Figure 7. It can be seen that the detection delay of LSTM-GAN on other test sets is significantly lower than that of RCS, ATA, and SVM, except that the detection delay of test 16 is higher than that of SVM. Therefore, compared with RCS, ATA, and SVM, LSTM-GAN performs better in terms of timeliness.

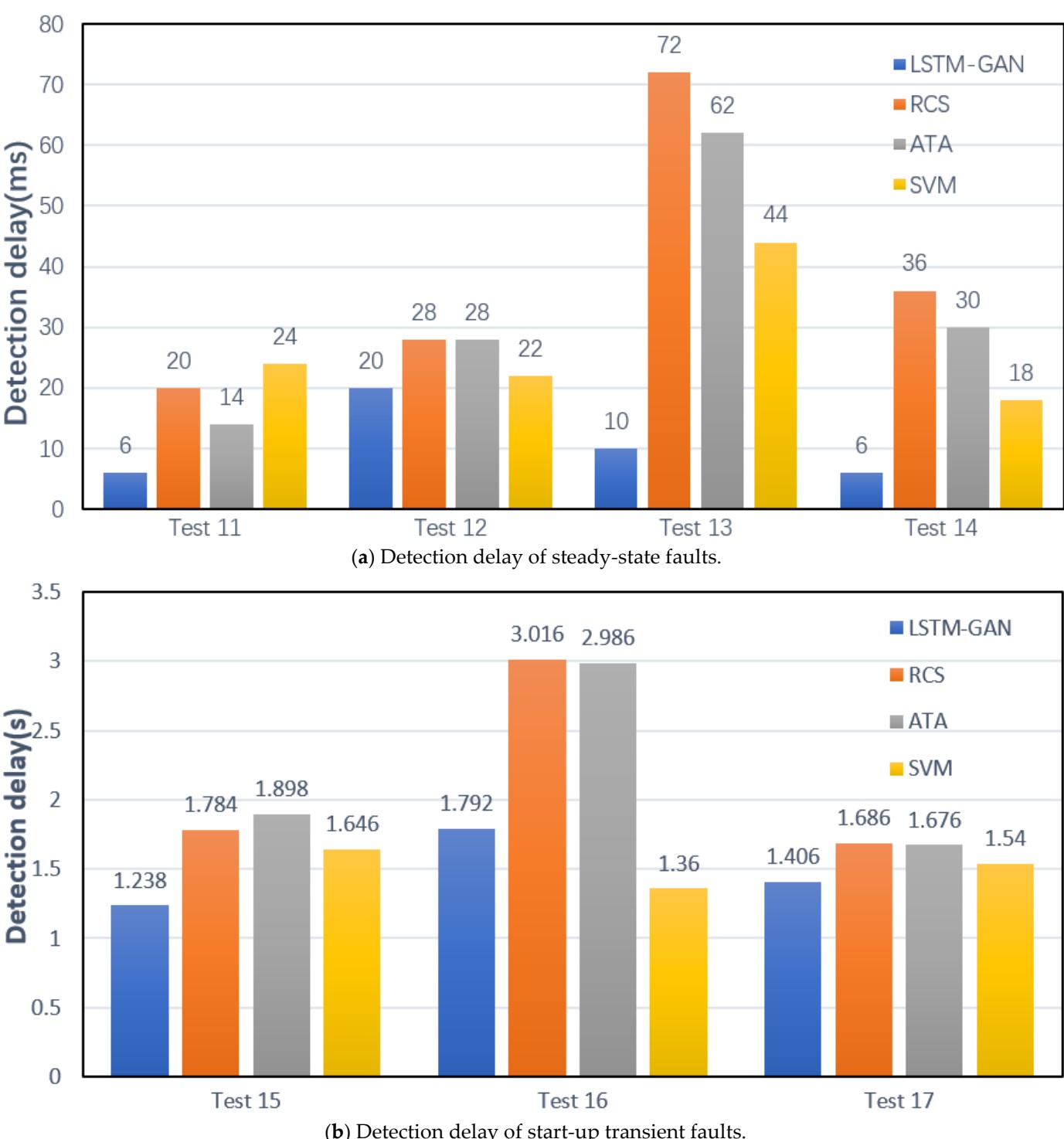

(**a**) Detection delay of steady-state faults.

(**b**) Detection delay of start-up transient faults.

**Figure 7.** Comparison between LSTM-GAN having a sample size of $10 \times 24$ and other methods.

*4.5. Fault Diagnosis Based on Generator*

Because of the complex LRE systems, the parameters in the startup process change significantly and rapidly, which makes it challenging to realize fault diagnosis. Therefore, this research only focuses on fault diagnosis for the steady-state process. According to the conclusion in Section 4.4, the LSTM-GAN having a sample size of $10 \times 24$ performs best. In addition, the data volume of the training set and test set is small, which significantly reduces the cost. Therefore, the model was selected for fault diagnosis.

We performed overlapping sampling according to the method described in Section 4.1, with a sampling step size of 1 and a window size of 10 to obtain the generator's input data. The test data are input into the generator, which outputs the predicted samples and calculates the absolute error between the predicted and the real value of each parameter. The absolute error for the *i*th parameter at a time is calculated as:

$$AE_i = |x_i - x_{ip}| \tag{11}$$

where $x_i$ and $x_{ip}$ are the real and the predicted values, respectively. There are many parameters and a wide range, so it is inappropriate to take the absolute error between the predicted value and the real value as the diagnostic index. The absolute error between the predicted value and the true value is standardized, and called the standardized absolute error (SAE). The calculation method of SAE for the *i*th parameter is calculated as:

$$SAE_i = \frac{AE_i - \mu_i}{\sigma_i} \tag{12}$$

where $\mu_i$ and $\sigma_i$ are the mean and standard deviation of $AE_i$ respectively. In order to prevent the introduction of test set information, the test set is standardized by using the mean and standard deviation of the absolute error between the predicted value and the real value of the training set. Finally, for each sample, the diagnostic index obtained is a 24-dimensional vector, **SAE**, i.e.,

$$\mathbf{SAE} = (SAE_1, SAE_2, \dots, SAE_{24}) \tag{13}$$

The corresponding fault diagnosis results are obtained by analyzing the diagnosis index. The faults of tests 11–14 are injection faults and the time of occurrence of the faults is clear, rapid, and complex. Therefore, the **SAE** of five consecutive samples after the injection fault was analyzed. The **SAE** of the five consecutive samples after the injection fault of tests 11–14 is shown in Figures 8–11.

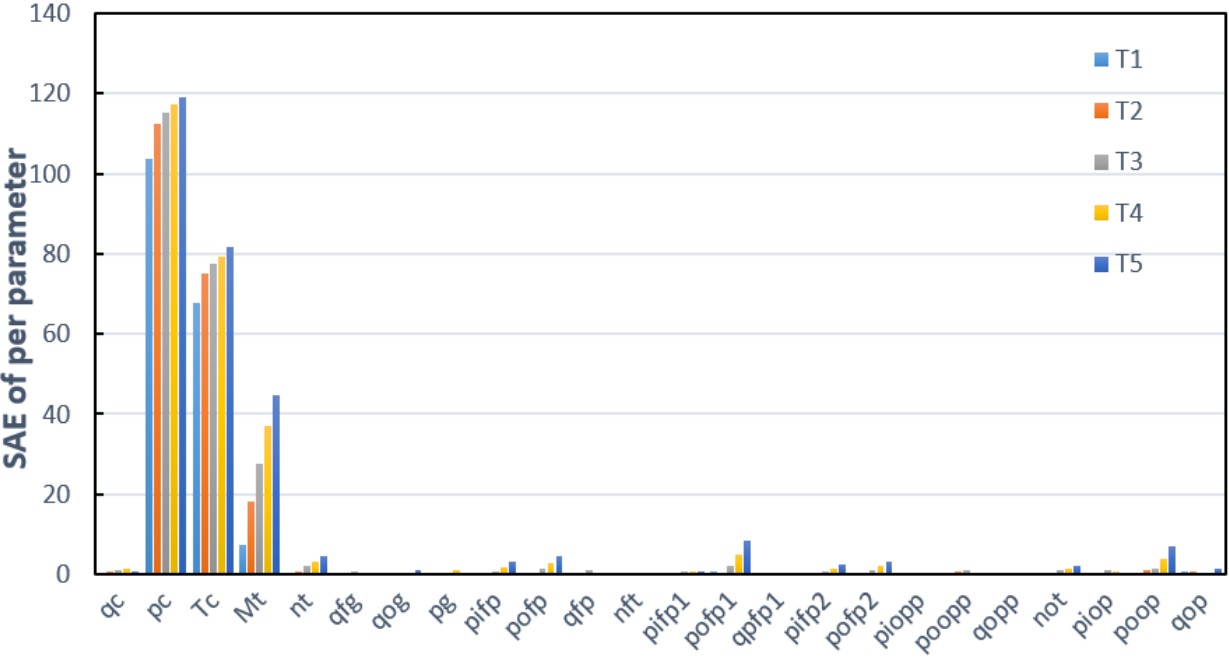

**Figure 8. SAE** of five consecutive samples after injection fault of test 11.

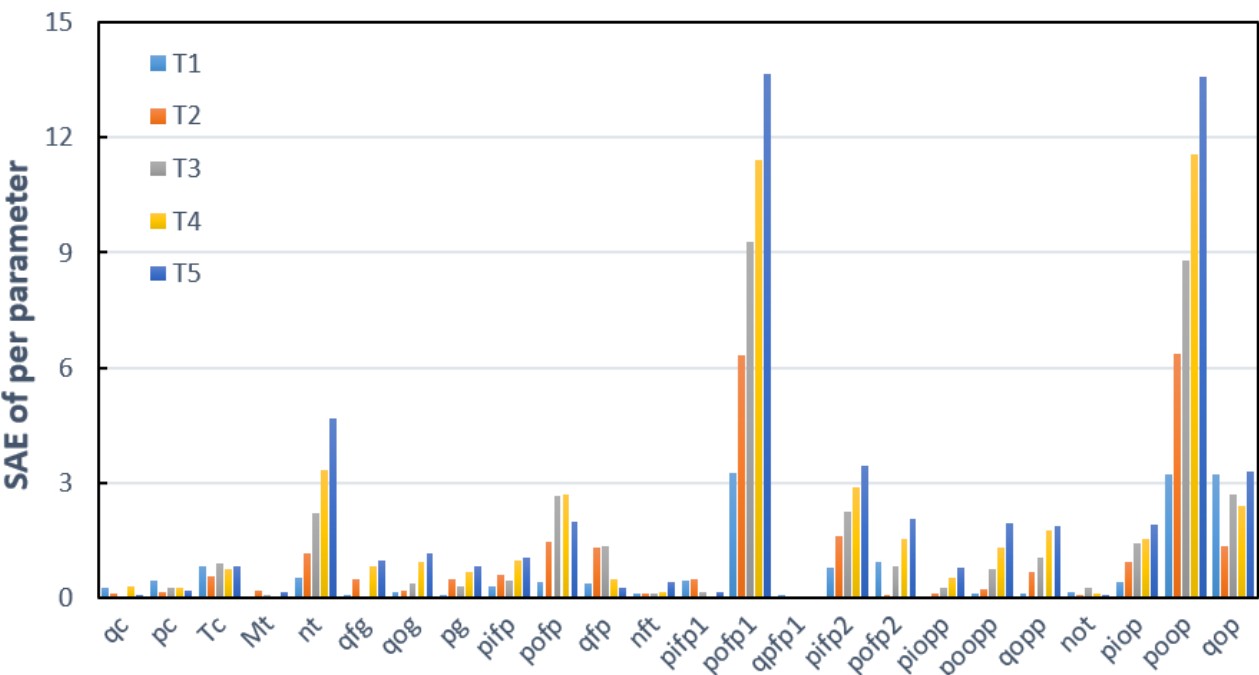

**Figure 9.** **SAE** of five consecutive samples after injection fault of test 12.

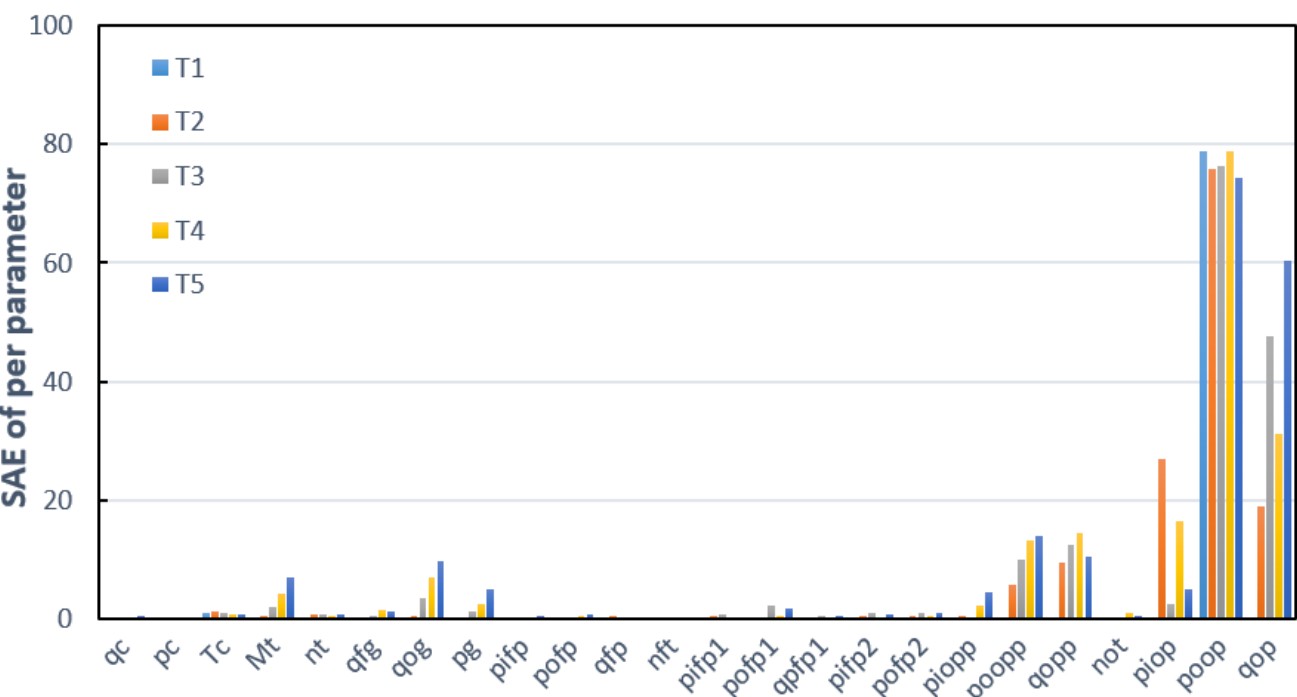

**Figure 10.** **SAE** of five consecutive samples after injection fault of test 13.

As shown in Figure 8, for test 11, the SAEs of *pc*, *Tc*, and *mt* increase rapidly and are much larger than that of other parameters. Assuming that the fault of test 11 is a combustion chamber fault, the SAE of *pc* and *Tc* first increases when the combustion chamber fails. Then, the failure affected the main turbine directly connected to it, causing the SAE of *mt* to increase. This process is consistent with the assumption. Therefore, the diagnosis result of test 11 is a combustion chamber fault, which is consistent with the injection fault.

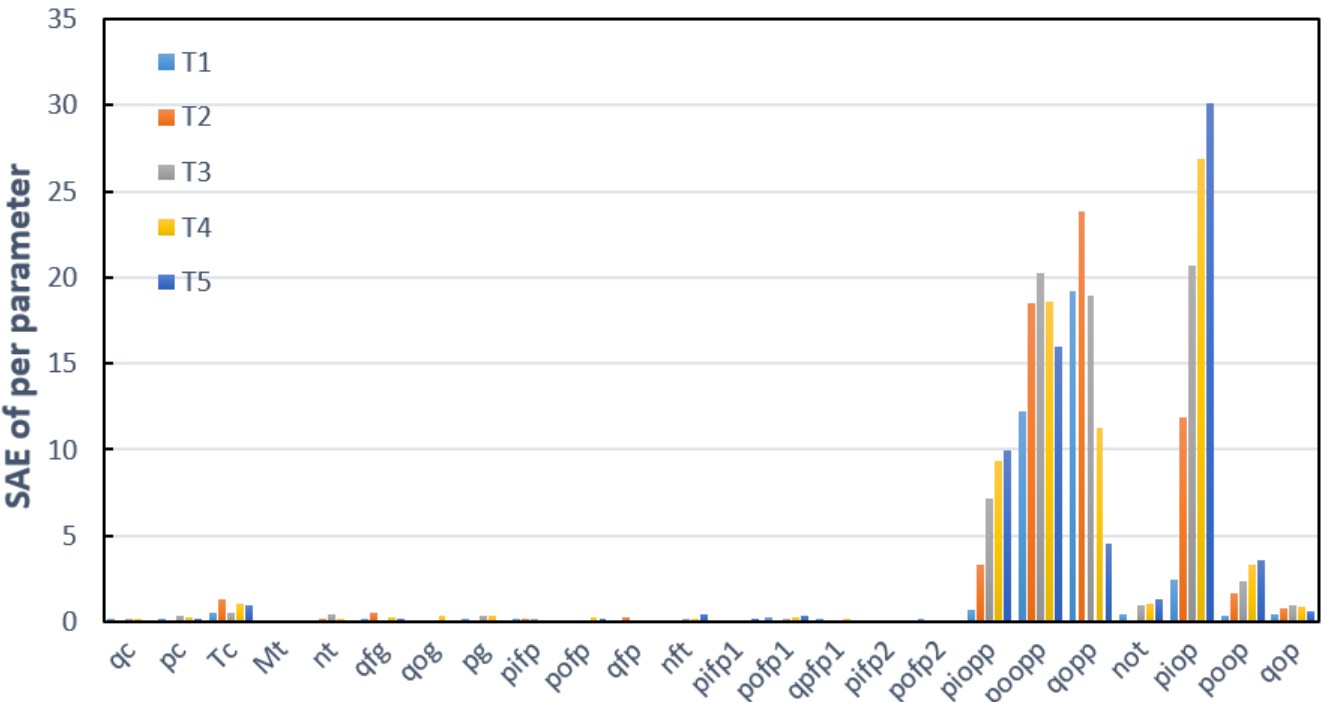

**Figure 11.** **SAE** of five consecutive samples after injection fault of test 14.

As shown in Figure 9, for test 12, the SAEs of *pofp1* and *poop* increase rapidly and are much larger than those of other parameters. In addition, *nt* is also significant. According to Figure 2, the main turbine, oxidant pump, and fuel primary pump are connected through a bearing. Therefore, the diagnosis result of test 12 is a bearing fault, which is consistent with the injection fault.

As shown in Figure 10, for test 13, the SAEs of *poop* and *qop* increase rapidly and are much larger than those of other parameters. Therefore, the fault diagnosis result of test 13 is an oxidizer pump fault, which is consistent with the injection fault.

As shown in Figure 11, for test 14, the SAEs of *piopp*, *poopp*, *qopp*, and *piop* are large. It is assumed that the fault of test 14 is the pipeline fault between the oxidant preloading pump and the oxidant pump. When the pipeline fails, the SAEs of the outlet pressure (*poopp*) and flow (*qopp*) of the oxidant preloading pump located upstream rapidly increase, and the SAE of the inlet pressure of the oxidant pump (*piop*) located downstream rapidly increases. This process is consistent with the assumption. Therefore, the diagnosis result of test 14 is the pipeline fault between the oxidant preloading pump and the oxidant pump, which is consistent with the injection fault.

## 5. Conclusions

This research proposes a fault detection and diagnosis method based on LSTM-GAN for the startup and steady-state process of a liquid oxygen/kerosene rocket engine. The performance of the proposed method was verified on the simulation data set. The corresponding results reveal that the proposed method will not produce false or missed alarms, and can realize the timely detection of faults in the LRE startup and steady-state processes. The fault components can be identified by analyzing the standardized absolute error between each parameter's predicted and true values, and fault diagnosis can be realized.

The advantage of our method is that it solely utilizes normal data to effectively train the LSTM-GAN model, overcoming the difficulty of insufficient fault data and realizing the detection and diagnosis of unknown faults. Of course, the proposed method also has some limitations. Due to the complexity of the startup process, it is challenging to realize fault diagnosis only by SAE. In order to address this problem, further research will be necessary in the future.

**Author Contributions:** Conceptualization, L.D. and Y.C.; methodology, L.D.; software, L.D.; validation, L.D. and Y.S.; formal analysis, L.D.; investigation, Y.C.; resources, Y.C.; data curation, Y.S.; writing—original draft preparation, L.D. and Y.S.; writing—review and editing, Y.C.; visualization, Y.S.; supervision, Y.C.; project administration, Y.C. All authors have read and agreed to the published version of the manuscript.

**Funding:** This research received no external funding.

**Institutional Review Board Statement:** Not applicable.

**Informed Consent Statement:** Not applicable.

**Data Availability Statement:** The data presented in this study are available on request from the corresponding author. The data are not publicly available due to privacy.

**Conflicts of Interest:** The authors declare no conflict of interest.

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
