# Peer review of "Fault Detection and Diagnosis for Liquid Rocket Engines Based on Long Short-Term Memory and Generative Adversarial Networks"

_aerospace, doi:10.3390/aerospace9080399_

Round 1
Reviewer 1 Report
This paper proposes a fault detection and diagnosis (FDD) method for a large LOX/kerosene rocket engine based on long short-term memory (LSTM) and generative adversarial networks (GAN). The study uses simulated data. The paper has a few minor deficiencies, but over its quality is acceptable and should be published.
An annotated copy is attached .
There are several suggested modifications.
1) There a "lot" of ACRONYMS TO SORT THOUGH IN THE TABLE. a list of Acronyms in the nomenclature would significantly increase clarity.
2) Figure 1 is not adequate. A functional block diagram describing the system layout would be significantly clearer.
3) Figure 2 needs to be expanded and embellished to depict/describe the very elaborate discussion of page 3.
4) In Section 1., Define/describe the "fault injection" process.
5) The data listed by Table 4 needs to be presented more compactly. A block diagram showing the test results for each parameter would be much better than the tabular listing, as presented. An inserted example in included in the annotated copy. The bar charts should list the nominal value foe each parameter, the injected errors for each tests, and the associated detection times.
6) In table 4, for each test, the authors average the detection times for different parameters, e.g. CHAMBER PRESSURE, Fuel-Flow rate, and combustor TEMPERATURE . Not the appropriate average .. should be across tests . you are averaging the detection times for different parameters .. apples and oranges!
7) Results are presented only for 3 tests, 11-13? Why? Authors should elaborate.
8) In the final section authors need to describe results in more detail .. are these results good? bad? how do these detection times compare to other methods? etc.
9) The conclusion is quite weak. It's like the authors "ran out of energy." Summarize your test results and then give them a quantitative or qualitative assessment. Avoid using words like"Straight-forward" .. describe what is meant here.

Author Response
Response to Reviewer 1 Comments
Point 1: There a "lot" of ACRONYMS TO SORT THOUGH IN THE TABLE. a list of Acronyms in the nomenclature would significantly increase clarity.
Response 1: I added acronyms to table 2.
Point 2: Figure 1 is not adequate. A functional block diagram describing the system layout would be significantly clearer.
Response 2: I replaced it with another figure, hoping it can meet the requirements.
Point 3: Figure 2 needs to be expanded and embellished to depict/describe the very elaborate discussion of page 3.
Response 3: I added some arrows to indicate the flow direction of the gas, making Figure 2 more in line with the text description.
Point 4: In Section 1., Define/describe the "fault injection" process.
Response 4: I rewrote the last paragraph of Section 1 and added the process of fault injection.
Point 5: The data listed by Table 4 needs to be presented more compactly. A block diagram showing the test results for each parameter would be much better than the tabular listing, as presented. An inserted example in included in the annotated copy. The bar charts should list the nominal value for each parameter, the injected errors for each test, and the associated detection times.
Response 5: Thank you very much for raising this point. I rewrote section 4.5, calculated the standardized absolute error in a period of time after fault injection, and clearly displayed the fault diagnosis results with a histogram.
Point 6: In table 4, for each test, the authors average the detection times for different parameters, e.g., CHAMBER PRESSURE, Fuel-Flow rate, and combustor TEMPERATURE. Not the appropriate average .. should be across tests. you are averaging the detection times for different parameters .. apples and oranges!
Response 6: Thank you for making this point. I have also considered whether it is appropriate to average these values, but I have not found a better way to make the results clearer. The rewritten Section 4.5 perfectly solves this problem.
Point 7:Results are presented only for 3 tests, 11-13? Why? Authors should elaborate.
Response 7: I explained the reasons in the first paragraph of Section 4.5.
Point 8: In the final section authors need to describe results in more detail .. are these results good? bad? how do these detection times compare to other methods? etc.
Response 8: I rewrote Section 4.4 and conclusion, adding a comparison with RCS, ATA and SVM.
Point 9: The conclusion is quite weak. It's like the authors "ran out of energy." Summarize your test results and then give them a quantitative or qualitative assessment. Avoid using words like"Straight-forward" .. describe what is meant here.
Response 9: Thanks for raising this point, I rewrote the conclusion.
Reviewer 2 Report
In the paper, a health monitoring system is presented to deliver diagnostic information and fault detection features for liquid rocket engines (LREs). In the introductory part of the paper the authors define reliability of the described system, giving an insight at the history of monitoring attempts and various deployments worldwide. Secondly, the research-related listing is given to present various approaches to the topic, what is a valuable part of the paper. This what is a drawback at this point is no clear informaton about novelty and contribution. Apart from that, the authors inform the readers about the simulator and training phases.
After presenting the simulator the authors describe their neural network (NN) to allow monitoring health statys of LREs. The question is, related to (1)-(5), why has the sigmoid function been selected. Please inform the readers why has this particular (see Sec. 3.3) NN been selected? Any prior trials with different structures, numbers of hidden layers, numbers of neurons per layers? Any quantitative information related to indices? Why has a L2-related performance index been selected? Are 'small' errors really neglectful (those below 1)? What is the methodology to select 'y' in (8), is it tuned on-line? What if the other values are selected?
Figure 6 - what is the reason for deterioration in training quality at epoch 3000?
Subsequently, the authors identify potential bottlenecks for timely monitoring of the LRE.
In the concluding part of the paper, the authors claim that their method proposes a fault detection and diagnosis method. None of these are presetend in the paper by solid results. No statistics, no plots, but time delay information. A vast experimental campaingn should be launched here to present solid statistics, including informaton about reliability rate of their approach, and what is most important - relating this to the existing methods.
Otherwise, this is a standalone paper, impossible to compare with the current approaches.
The paper needs a second attemt in its submission, after all the errors are removed.
Author Response
Response to Reviewer 2 Comments
Point 1: This what is a drawback at this point is no clear information about novelty and contribution.
Response 1: I rewrote the last paragraph of the first part and added a description of innovation and contribution.
Point 2: The question is, related to (1)-(5), why has the sigmoid function been selected?
Response 2: Sorry, due to my negligence, formula (3) is wrong. In fact, I chose tanh function. I modified this part, added the formulas of sigmoid function and tanh function, and explained the functions of sigmoid function and tanh function. In addition, I modified formulas (1) - (6) and related description words to make the formula clearer and the description clearer.
Point 3: Please inform the readers why has this particular (see Sec. 3.3) NN been selected?
Response 3: I rewrote the last paragraph of Section 1 to explain the reasons for using LSTM-GAN.
Point 4: Any prior trials with different structures, numbers of hidden layers, numbers of neurons per layers? Any quantitative information related to indices?
Response 4: In previous years' papers on the application of LSTM in the field of FDD, many explorations have been made on the number and layers of LSTM neurons, and researchers have explored the appropriate number and layers of neurons in this field. In recent years, most of the papers use LSTM layers with 1-3 layers, 128 or 256 neurons. I used the most used two-layer LSTM and 128 neurons, and achieved good results, so I didn't explore related aspects.
Point 5: Why has a L2-related performance index been selected?
Response 5: The generator built by LSTM solves the regression problem, and therefore the MSE loss function is selected. In addition, since the generator needs to consider the discriminator’s result when up-dating the parameters, the generator’s loss function is obtained by adding the two parts to obtain the average value. To preserve consistency with the generator, we select the MSE loss function for the discriminator.
Point 6: Are 'small' errors really neglectful (those below 1)?
Response 6: I'm sorry, I didn't understand what you meant. I would be grateful if you could explain.
Point 7: What is the methodology to select 'y' in (8), is it tuned on-line? What if the other values are selected?
Response 7: It is not adjusted online. 'x' appeared in (7), so I'm worried that using 'x' again will cause ambiguity. After your inquiry, I found that my worry was unnecessary, so I changed 'y' to 'x'.
Point 8: Figure 6 - what is the reason for deterioration in training quality at epoch 3000?
Response 8: I stopped the training of the model in advance and got better results. Thank you very much.
Point 9: In the concluding part of the paper, the authors claim that their method proposes a fault detection and diagnosis method. None of these are presented in the paper by solid results. No statistics, no plots, but time delay information. A vast experimental campaign should be launched here to present solid statistics, including information about reliability rate of their approach, and what is most important - relating this to the existing methods.
Response 9: Thanks for raising this point. I rewrote the conclusion and added the comparison with RCS, ATA and SVM.
Round 2
Reviewer 2 Report
The paper is acceptable for publication after its previous revision. Good luck with the review process.